# Self-Augmented Preference Optimization: Off-Policy Paradigms for Language Model Alignment

## Abstract

Traditional language model alignment methods, such as Direct Preference Optimization (DPO), are limited by their dependence on static, pre-collected paired preference data, which hampers their adaptability and practical applicability. To overcome this limitation, we introduce Self-Augmented Preference Optimization (SAPO), an effective and scalable training paradigm that does not require existing paired data. Building on the self-play concept, which autonomously generates negative responses, we further incorporate an off-policy learning pipeline to enhance data exploration and exploitation. Specifically, we employ an Exponential Moving Average (EMA) model in conjunction with a replay buffer to enable dynamic updates of response segments, effectively integrating real-time feedback with insights from historical data. Our comprehensive evaluations of the LLaMA3-8B and Mistral-7B models across benchmarks—including the Open LLM Leaderboard, IFEval, AlpacaEval 2.0, and MT-Bench—demonstrate that SAPO matches or surpasses established offline contrastive baselines, such as DPO and Odds Ratio Preference Optimization, and outperforms offline self-play methods like SPIN.

## 1 Introduction

In the rapidly evolving field of artificial intelligence, aligning Large Language Models (LLMs) with human preferences has emerged as a critical area of research (Agrawal et al., 2023; Shi et al., 2023; Kadavath et al., 2022; Liang et al., 2021; Sheng et al., 2019; Christiano et al., 2017). Classical methods, such as Reinforcement Learning (RL) from Human Feedback (RLHF) (Ziegler et al., 2019; Ouyang et al., 2022), have progressed by training models to optimize responses via a reward model that reflects human preferences. However, the necessity of a separate reward model introduces additional complexity and computational demands. To streamline this, Direct Preference Optimization (DPO) (Rafailov et al., 2023) directly utilizes preference data to optimize language models, eliminating the need for an auxiliary reward model. Odds Ratio Preference Optimization (ORPO) (Hong et al., 2024) further streamlines the alignment process by removing the reference model entirely. ORPO employs an odds ratio to directly evaluate preferences between different responses during Supervised Fine-Tuning (SFT), thus simplifying the alignment process. However, despite enhancements with DPO, ORPO, and many other offline contrastive preference learning algorithms (Hong et al., 2024; Ethayarajh et al., 2024; Zhao et al., 2023), their reliance on static, pre-collected preference datasets poses challenges, especially in sensitive domains where privacy concerns and the scarcity of expert input limit adaptability and application scope.

The Self-Play Fine-Tuning (SPIN) method (Chen et al., 2024) tackles the challenge of data collection by using a self-play approach, where the model autonomously generates its own responses to serve as rejected inputs. This strategy enables SPIN to function with minimal data requirements—only requiring prompts and selected responses—thereby alleviating the difficulty of gathering paired preference datasets. However, SPIN's methodology comes with significant limitations. Its primary drawback is the reliance on offline, pre-generated responses, which hampers the model's ability to dynamically adjust training data in real-time. Additionally, this dependency necessitates a rigid training procedure, where complete data generation must precede the start of training, introducing significant delays.

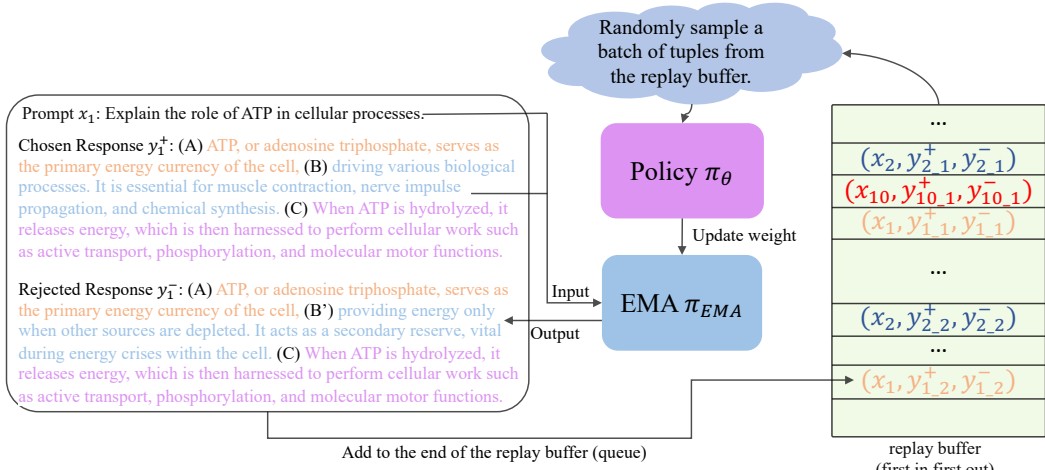

Figure 1: Given a prompt $x_1$ and chosen response $y_1^+$. This response is segmented into $A$, $B$, and $C$. Using the prompt with segment $A$, the EMA model generates a new segment $B'$. Together, segments $A$, $B'$, and $C$ form the rejected response $y_1^-$, which is appended to the replay buffer. Random tuples are sampled from this buffer to train the policy network, subsequently updating the EMA weights.

Recent work has underscored the significance of online training in enhancing the alignment performance of Large Language Models (LLMs) (Guo et al., 2024; Rosset et al., 2024). However, these methods largely rely on reward models or powerful teacher models like GPT-4 to provide guidance signals. We introduce Self-Augmented Preference Optimization (SAPO), as depicted in Figure 1, a general self-play preference learning algorithm without depending on any external reward models or teacher models. We derive the motivation for SAPO from the principles of off-policy RL training. (Lillicrap et al., 2015; Haarnoja et al., 2018; Wang et al., 2022). SAPO consists of three main components: the current policy, an Exponential Moving Average (EMA) model of the current policy, and a first-in-first-out replay buffer.

The learning of SAPO involves two stages at each iteration: sampling and training. During the sampling stage, the EMA model is used to generate responses, creating self-augmented rejected samples. Both the original responses and these generated samples, along with the prompts, are stored in the replay buffer. In the training stage, we randomly select a batch of tuples from the replay buffer and employ typical preference learning methods, such as DPO (Rafailov et al., 2023) and ORPO (Hong et al., 2024), to train the current policy. After training, we update the EMA model at a fixed rate. These two stages work in tandem to incrementally improve the policy. By using the EMA model and replay buffer, we reduce the impact of volatility from any single training iteration and ensure more consistent learning signals. The progression of training data within the replay buffer adopts principles akin to curriculum learning (Xu et al., 2020; Wang et al., 2024b; Pattnaik et al., 2024), starting with simpler training pairs and gradually incorporating more complex training samples, allowing the model to build competency progressively.

SAPO employs a teacher-forcing segment-level supervision strategy, truncating a chosen response at a random point to create a supervision segment. As illustrated in Figure 1, with the prompt "Explain the role of ATP in cellular processes," a chosen response segment $B$ might be "When ATP is hydrolyzed, it releases energy, which is then harnessed to perform cellular work." Conversely, the EMA model might generate a less accurate rejected segment $B'$, such as "providing energy only when other sources are depleted," positioning ATP erroneously as a secondary reserve. By focusing on generating tailored segments instead of entire rejected responses from scratch, SAPO is more likely to produce meaningful outputs. This method facilitates more tailored adjustments to the training data.

Experimental evaluations across four benchmarks—Open LLM Leaderboard (Beeching et al., 2023), which tests question answering ability; IFEval (Zhou et al., 2023), measuring instruction-following ability; MT-Bench (Zheng et al., 2024) and AlpacaEval 2.0 (Dubois et al., 2024), assessing conversational ability-demonstrate that SAPO can either match or exceed the performance of existing offline contrastive learning methods like DPO (Rafailov et al., 2023) and ORPO (Hong et al., 2024), despite

our method solely utilizing chosen responses. Furthermore, SAPO outperform purely offline self-play methods such as SPIN (Chen et al., 2024), which require longer training times.

## 2 PRELIMINARIES

### 2.1 DEFINITION OF LEARNING PARADIGMS IN LLM ALIGNMENT

For ease of understaning RL concepts in LLMs, we provide the following definitions:

- **Offline Learning:** Offline learning involves the LLM being trained on a pre-collected dataset without further interaction with additional reward models or acquiring new human annotated data during training.
- **On-Policy Learning:** This learning approach ensures that the training data is generated and utilized under the current policy of the LLM. It implies that the model is refined using data derived from the decisions it would make under its current strategy.
- **Off-Policy Learning:** Off-policy learning enables the use of data generated from a different policy than the one currently being trained. This approach allows the model to leverage data from past policies or alternative strategies, providing a broader range of experiences for the model to learn from, which may not necessarily align with its current operational policy.

### 2.2 OFFLINE OFF-POLICY CONTRASTIVE PREFERENCE LEARNING ALGORITHMS

DPO (Rafailov et al., 2023) is designed to optimize a policy $\pi_\theta$, based on a reference model $\pi_{\text{ref}}$ that is typically the SFT baseline. DPO utilizes an off-policy, offline training approach, employing a pre-collected dataset of triplets $(x, y^+, y^-)$ to enable preference learning. Within this dataset, $x$ serves as the input prompt, $y^+$ as the chosen response, and $y^-$ as the rejected response. The DPO loss function is formulated as follows:

$$\mathcal{L}_{\text{DPO}}(\pi_\theta; \pi_{\text{ref}}) = -\mathbb{E}_{(x,y^+,y^-)\sim\mathcal{D}}\left[\log\sigma\left(\beta\log\frac{\pi_\theta(y^+|x)}{\pi_{\text{ref}}(y^+|x)} - \beta\log\frac{\pi_\theta(y^-|x)}{\pi_{\text{ref}}(y^-|x)}\right)\right] \quad (1)$$

Here, $\beta$ is a hyperparameter that controls the degree of KL regularization.

ORPO (Hong et al., 2024) provides a unique method for optimizing policy models without the need for a reference model. By employing log odds ratios, ORPO directly contrasts favored and disfavored responses during SFT, simplifying model alignment. Utilizing the odds defined as:

$$\text{odds}(y^+|x) = \frac{\pi_\theta(y^+|x)}{1 - \pi_\theta(y^+|x)}, \quad \text{odds}(y^-|x) = \frac{\pi_\theta(y^-|x)}{1 - \pi_\theta(y^-|x)} \quad (2)$$

the ORPO algorithm compute the log odds ratio, effectively balancing the enhancement and penalization of responses, as:

$$\mathcal{L}_{\text{ORPO}} = \mathbb{E}_{(x,y^+,y^-)\sim\mathcal{D}}\left[\mathcal{L}_{\text{SFT}} - \lambda \cdot \log\sigma\left(\log\frac{\text{odds}(y^+|x)}{\text{odds}(y^-|x)}\right)\right] \quad (3)$$

Here, $\mathcal{L}_{\text{SFT}}$ is the supervised fine-tuning loss aimed at maximizing the likelihood of generating the chosen responses, and $\lambda$ is a hyperparameter that weights the relative importance of the odds ratio term in the overall loss function.

### 2.3 SELF-PLAY FINE-TUNING

The Self-Play Fine-Tuning (SPIN) method (Chen et al., 2024) introduces a self-play mechanism that reduces the reliance on pre-collected paired preference data. Unlike traditional approaches, SPIN necessitates only annotated chosen data, as it autonomously generates its own rejected data from its previous iterations. SPIN then utilizes existing RLHF methods, *e.g.*, DPO, to iteratively refine model responses by discerning these self-generated responses from those obtained from human-annotated data, a process inspired by principles of game theory (Samuel, 1959). While SPIN effectively eliminates the reliance on offline contrastive learning's need for paired data by autonomously generating rejected responses, it utilizes pre-generated training data from the model's prior iterations. This data remains unchanged throughout subsequent training cycles, which may limit the model's ability to adapt to new information or evolving training conditions.

## 3 SELF-AUGMENTED PREFERENCE OPTIMIZATION

Recent advancements in offline contrastive preference learning algorithms (Rafailov et al., 2023; Ethayarajh et al., 2024; Hong et al., 2024) have shown promising results in aligning LLMs with human preferences. However, these methods typically rely on pre-collected paired preference datasets. Our goal is to create a robust and efficient self-augmented algorithm that eliminates the requirement for paired data. This new algorithm autonomously generates high-quality rejected responses which, combined with chosen responses from SFT datasets, form the necessary pairs for preference learning. Recent initiatives like SPIN (Chen et al., 2024) utilize an iterative training method involving self-play, where sampling and training occur in separate phases. Initially, rejected responses are sampled across the entire dataset, followed by a distinct training phase. This iterative paradigm tends to be slow because it alternates between sampling for the entire dataset and training phases. As the training progresses, the effectiveness of the generated preference learning pairs may diminish. This is because the model continues to evolve, whereas the sampled dataset remains static for each iteration and cannot adapt to the latest model updates.

Instead, we start from a standard off-policy RL framework. To efficiently sample high-quality preference pairs, we introduce segment-level supervision. This approach involves replacing a segment of rejected responses with outputs generated by LLM, thereby naturally creating challenging negative sample pairs through targeted local modifications. Furthermore, we integrate an EMA model and a replay buffer—techniques well-established in RL—to facilitate standard off-policy training. This strategy ensures timely feedback and updates to the training data, operating independently of external feedback mechanisms. The implementation of SAPO is summarized in Algorithm 1, and a comparison with on-policy training is provided in Section 5.3.

**Segment-Level Supervision in SAPO.** Unlike SPIN (Chen et al., 2024), which generates rejected responses from scratch, SAPO utilizes a teacher-forcing segment-level supervision method to refine the learning process. Consider a SFT dataset $\mathcal{D}$ consisting of tuples $(x, y^+)$, where $x$ is a prompt and $y^+$ is the selected response. During each training iteration, the model randomly selects a truncation point within each response $y^+$, defining segment $B$ of length $N_{\text{seg}}$ starting from this point. Segment $A$ comprises the tokens preceding $B$, and segment $C$ includes the tokens following $B$. The original response $y^+$ can thus be expressed as:

$$y^+ = A \oplus B \oplus C \tag{4}$$

The model attempts to regenerate $B$ as $B'$ based on the prompt and segment $A$. For continuity and to maintain the contextual integrity of the response, segment $C$ is concatenated, resulting in:

$$y^- = A \oplus B' \oplus C, \tag{5}$$

where $\oplus$ denotes concatenation. This segmentation strategy not only improves supervision granularity by focusing on specific response segments – by regenerating only the middle segment $B'$, the model can concentrate its learning efforts on specific parts of the response that may be problematic or less accurate. In addition, sampling segments is more time efficient than sampling the complete sentences.

**Off-Policy Learning Setting.** Inspired by off-policy RL training methods (Lillicrap et al., 2015; Haarnoja et al., 2018; Wang et al., 2022), the proposed SAPO framework incorporates several fundamental components: the current policy, an EMA model of this policy, and a first-in-first-out replay buffer. In this context, the replay buffer $\mathcal{B}$ plays a crucial role, mirroring curriculum learning principles (Xu et al., 2020; Wang et al., 2024b; Pattnaik et al., 2024). The recent Curry-DPO study (Pattnaik et al., 2024) also highlights the effectiveness of sequencing preference pairs from simpler to more complex throughout training, which gradually increases task complexity and enhances learning efficiency. This approach prevents the model from being overwhelmed by challenging tasks at early stages, thus improving learning outcomes and enhancing model robustness. However, the Curry-DPO approach requires an additional reward model to manually order training pairs, which can introduce biases and require extra resources. Our method automates this process, naturally achieving a curriculum learning effect and reducing dependency on supplementary models.

Initially, our replay buffer is populated with simple training pairs—where the rejected responses $(y^-)$ are distinctly different from the chosen responses $(y^+)$, and these responses are generated by the early iterations of the model. As the model evolves and its capacity for generating better

---

**Algorithm 1** Self-Augmented Preference Optimization (SAPO)

---

1: **Input:** Dataset with prompts and responses, base model $\pi_\theta$, total number of iterations $T$, learning rate $lr$, EMA coefficient $\alpha$
2: Initialize replay buffer $\mathcal{B}$
3: Initialize EMA model parameters $\theta_{\text{EMA}}$ with $\theta$
4: **for** each iteration $i$ from 1 to $T$ **do**
5:    **# Sampling Stage**
6:    Sample a mini-batch of $(x, y^+)$ tuples from the dataset, each batch containing $N$ samples.
7:    **for** each $(x, y^+)$ in the batch **do**
8:       Randomly truncate $y^+$ to obtain segments $A$, $B$, and $C$.
9:       Combine $x$ with segments $A$ and $B$ as input to the EMA model to generates segment $B'$.
10:      Concatenate $A$, $B'$, and $C$ to form the rejected response $y^-$.
11:      Store $(x, y^+, y^-)$ in the replay buffer $\mathcal{B}$.
12:    **end for**
13:    **# Training Stage**
14:    Sample a mini-batch of tuples $(x, y^+, y^-)$ from $\mathcal{B}$
15:    Compute loss $\mathcal{L}$ using DPO/ORPO formulas (Eq. 1 and Eq. 3) based on tuples $(x, y^+, y^-)$
16:    Update policy parameters $\theta$ using gradient descent: $\theta \leftarrow \theta - lr\nabla_\theta\mathcal{L}(x, y^+, y^-, \theta)$
17:    Update EMA model parameters: $\theta_{\text{EMA}} \leftarrow \alpha\theta_{\text{EMA}} + (1 - \alpha)\theta$
18: **end for**

---

responses increases, the quality of newly generated $y^-$ responses also improves. These $y^-$ responses are not generated from the current fine-tuning policy model but from the EMA model, denoted as $\pi_{\text{EMA}}$. This mechanism is designed to stabilize training by utilizing a less variable model state to generate responses, thereby reducing the impact of any single training iteration's volatility on the overall learning process. Our replay buffer operates as a queue, adhering to the FIFO (First In, First Out) principle, which facilitates the gradual replacement of simpler, initial training examples with more complex ones, embodying offline learning by reusing accumulated past data. This progression naturally mirrors curriculum learning, where tasks that start simply become progressively more challenging, thereby enhancing the model's training stability. Additionally, we have implemented a sampling mechanism within the replay buffer where each entry is equipped with a counter that increases each time that entry is sampled. This setup ensures that the sampling weight for each entry becomes inversely proportional to its frequency of use, preventing over-sampling of certain data and promoting a more even distribution of sample usage. This balanced approach is crucial for ensuring that both historical insights and fresh perspectives are consistently integrated into the model's learning, thereby enhancing the robustness of the training process.

The EMA model further supports this off-policy learning setting by stabilizing the learning process through the averaging of policy parameters $\theta$, updated as:

$$\theta_{\text{EMA}} \leftarrow \alpha\theta_{\text{EMA}} + (1 - \alpha)\theta, \tag{6}$$

where $\alpha$ is a decay factor. This approach exemplifies the off-policy nature of the learning process, as the data used is not directly generated from the currently fine-tuned policy. Our experiments have shown that this off-policy method yields better results compared to directly using on-policy data for model fine-tuning. Such a comprehensive strategy ensures that SAPO remains adaptive and effective, refining the model's response generation capabilities throughout its training process.

## 4 RELATED WORK

### 4.1 REINFORCEMENT LEARNING FROM HUMAN FEEDBACK (RLHF)

RLHF Methods can be categorized into two primary approaches: reward-based methods and reward-free methods. Reward-based methods like Proximal Policy Optimization (PPO) (Schulman et al., 2017) utilize a trained reward model (Ziegler et al., 1909; Stiennon et al., 2020; Ouyang et al., 2022) to provide feedback signals for online RL algorithms. The training of multiple models (policy, reward, and advantage model) increases computational demands and can lead to instability during the training process (Gao et al., 2023; Wang et al., 2024a). In contrast, reward-free methods simplify

the training process by eliminating the need for a separate reward model. DPO (Rafailov et al., 2023) integrates the reward modeling stage directly into the preference learning stage. This method, based on a closed-form solution derived from the Bradley-Terry model Bradley & Terry (1952), is noted for its efficiency and stability. (Zhao et al., 2023) introduce Sequence Likelihood Calibration with Human Feedback (SLiC-HF), which employs a contrastive ranking calibration loss combined with a regularization loss to refine the scoring of responses. The Kahneman-Tversky Optimization (KTO) algorithm (Ethayarajh et al., 2024) leverages human utility principles to optimize language models using unpaired data, moving away from the dependency on pairwise preference datasets. Relative Preference Optimization (RPO) (Yin et al., 2024) utilizes a contrastive weighting mechanism that evaluates preferences across not only identical but also semantically similar prompts, allowing for both paired and unpaired data scenarios. Recently, the ORPO algorithm (Hong et al., 2024) simplifies preference alignment by integrating supervised fine-tuning and preference optimization into a single training stage without requiring a reference model. However, a major issue with offline contrastive preference learning approaches is their dependence on static, pre-collected paired preference datasets, typically involving a single optimization procedure. This reliance can lead to a distribution shift between the offline training data and the fine-tuned model, potentially impacting the model's effectiveness and adaptability.

## 4.2 ITERATIVE FINE-TUNING LLMs

Iterative fine-tuning enhances language models by using outputs from the model itself or external models as inputs for subsequent training iterations, aiming to improve performance from each training cycle. A family of iterative methods (Li et al., 2023; Gulcehre et al., 2023; Hu et al., 2023; Mukobi et al., 2023) involves continuously refining language models by supervised fine-tuning models on carefully selected, preferred responses. This iterative approach is further applied within the DPO framework, as demonstrated by a number of recent works (Xu et al., 2023; Xiong et al., 2024; Yuan et al., 2024; Chen et al., 2024). Utilizing iterative DPO-type training, updated models generate new preference pairs at each iteration (Xu et al., 2023; Xiong et al., 2024; Guo et al., 2024). These pairs are then scored using feedback from additional reward models or human evaluations. Yuan et al. (2024) introduce Self-Rewarding Language Models, where the model annotates its own responses. Integrated into the iterative DPO framework, this allows the model to autonomously generate and assess preference pairs, streamlining fine-tuning by reducing external feedback reliance. However, this self-judging approach heavily relies on the model's own evaluative capabilities, making it more suitable for larger parameter language models. The SPIN algorithm (Chen et al., 2024) uses a self-play framework for iterative DPO-style fine-tuning, labeling human-generated responses as winners and model-generated ones as losers. However, SPIN generates datasets for the next cycle offline, which limits the incorporation of fresh outputs from updated models. Additionally, its reliance on offline learning can lead to a shift problem as the fine-tuned model increasingly diverges from the one used to generate the preference dataset. To address these challenges, we have integrated real-time data sampling within the self-play framework, facilitating immediate updates to the training data. Some recent works (Rosset et al., 2024; Wu et al., 2024) have also highlighted the importance of online iteration in preference learning. Unlike these approaches, which often rely on additional reward models or more advanced teacher models like GPT-4, our method seeks to develop a general and effective self-augmented algorithm that functions independently of external supervision.

Some research focuses on domain-specific applications to self-improve language models. For instance, Lee et al. (2024) target low-data regime tasks, optimizing language models with limited initial datasets. Other works concentrate on enhancing reasoning capabilities, as seen in Pang et al. (2024). In contrast, our research focus on enhancing language models for general instruction-following tasks.

## 5 EXPERIMENTS

### 5.1 EXPERIMENTAL SETUP.

**Baselines.** We compare two offline contrastive preference learning algorithms: DPO (Rafailov et al., 2023) and ORPO (Hong et al., 2024). Additionally, we adapt the SPIN algorithm (Chen et al., 2024) for both DPO and ORPO. We have implemented two versions of our SAPO algorithm, each utilizing the loss functions from DPO and ORPO, respectively. Both DPO and ORPO require paired data for training, doubling the dataset size compared to SPIN and SAPO. Notably, SPIN's sequential process

of generating and training on data not only introduces considerable delays, as training cannot begin until the entire dataset has been generated, but also adds complexity to the workflow. This results in higher latency in model readiness when compared to SAPO, which utilizes a more streamlined and efficient approach.

**Datasets and Base Models.** We utilize the Distilabel-Capybara dataset[1], a multi-turn dialogue preference dataset comprising 7.6k entries, designed to enhance the conversational abilities of open-source LLMs. Each entry consists of multiple dialogue turns between a user and an assistant, with only the final message from the assistant considered as a response, while the preceding interactions serve as the prompt. Responses are generated by various LLMs, and then assessed using gpt-4-turbo. Although the dataset is relatively small in size, it is of high quality. For contrastive offline preference learning algorithms like DPO and ORPO, both chosen and rejected responses are required as training data. In contrast, for SPIN and our SAPO, only the prompts and chosen responses from the Distilabel-Capybara dataset are necessary.

We experiment with two types of models, Mistral and LLaMA, for DPO and ORPO-based algorithms. For DPO-based models utilizing the Mistral architecture, we employ mistral-7b-zephyr-sft[2] as the base model, which undergoes supervised fine-tuning on the Deita 10k dataset (Liu et al., 2023). For the DPO-based LLaMA model, we use Meta-LLaMA-3-8B as the base model, following the Mistral SFT protocol by conducting supervised fine-tuning on the Deita 10k dataset to produce the llama-3-8b-sft model. For the ORPO algorithm, our base models include Mistral-7B-v0.1 and Meta-LLaMA-3-8B. These models are then utilized in various preference learning algorithms to achieve preference alignment.

**Training details.** All experiments were conducted using 8 Nvidia H100 GPUs. For the Distilabel-Capybara dataset, the maximum prompt length was set to 1792, and the combined length of the prompt and response was capped at 2048. For the AlpacaEval 2.0 evaluation, we employed the officially recommended weighted_alpaca_eval_gpt4_turbo as the annotator.

Before implementing the Direct Preference Optimization (DPO) algorithm, the base models underwent Supervised Fine-tuning (SFT). The Mistral-7B model used a pretrained version available on Huggingface under wandb/mistral-7b-zephyr-sft. For the LLaMA-3-8B model, we replicated the Mistral's training setup, with a learning rate set to 2.0e-05. The models were fine-tuned over three epochs on the Deita 10k dataset, known for its high quality and suitability for SFT. We employed a cosine learning rate scheduler and set the maximum sequence length at 2048 tokens.

For DPO (Rafailov et al., 2023), the KL regularization coefficient $\beta$ was set at 0.1 with a learning rate of 5.0e-7, using the rmsprop optimizer. The training spanned three epochs.

For ORPO (Hong et al., 2024), we adhered to the default settings used on the Distilabel-Capybara dataset. The odds ratio weight coefficient $\lambda$ was set at 0.05, with a learning rate of 5.0e-6, using the AdamW optimizer (Loshchilov & Hutter, 2017). The ORPO training was conducted for three epochs on the Mistral model and four epochs on the LLaMA model.

SPIN (Chen et al., 2024) was configured similarly to DPO and ORPO, running across four iterations, each consisting of three epochs. The learning rate was 5.0e-7 for the first two iterations and reduced to 1e-7 for the remaining two. The $\beta$ coefficient was increased to 5.0 for the final iteration.

**Evaluation Benchmarks.** Following previous studies (Hong et al., 2024; Chen et al., 2024), we assess the model performance using four established benchmarks, including the Open LLM Leaderboard (Beeching et al., 2023), IFEval (Zhou et al., 2023), MT-Bench (Zheng et al., 2024), and AlpacaEval 2.0 (Dubois et al., 2024). These benchmarks enable us to comprehensively evaluate our approach and baseline methods across various aspects, including question answering, instruction-following, and conversational ability.

- **Open LLM Leaderboard (Beeching et al., 2023):** A comprehensive benchmark suite aggregating six popular datasets: ARC (Clark et al., 2018), GSM8K (Cobbe et al., 2021), HellaSwag (Zellers et al., 2019), MMLU (Hendrycks et al., 2020), TruthfulQA (Lin et al.,

---

[1]https://huggingface.co/datasets/argilla/distilabel-capybara-dpo-7k-binarized
[2]https://huggingface.co/wandb/mistral-7b-zephyr-sft

2021), and Winogrande (Sakaguchi et al., 2021). This leaderboard assesses diverse aspects of language model performance including reasoning, language understanding, and problem-solving through few-shot prompting on these test sets.

- **IFEval (Zhou et al., 2023):** IFEval benchmark evaluates language models on their ability to follow instructions, featuring 541 prompts with verifiable directives such as length constraints and specific formats. This benchmark assesses models using 25 different types of instructions in a zero-shot evaluation, focusing on the accuracy and compliance of models in executing detailed instructions.
- **MT-Bench (Zheng et al., 2024):** MT-Bench tests language models with 80 multi-turn questions across domains like writing and coding. Each question set challenges models to maintain context over two turns. GPT-4 (OpenAI, 2023) rates responses from 1 to 10, and the overall score is averaged to evaluate the model's conversational skill and understanding across subjects.
- **AlpacaEval 2.0 (Dubois et al., 2024):** AlpacaEval 2.0 employs a dataset of 805 input prompts for a win-rate comparison against GPT-4-Turbo. This benchmark focuses on evaluating response quality through a win-rate mechanism that has been enhanced in its latest version to control for length bias. This adjustment ensures that evaluations accurately reflect the substantive quality of the responses rather than their length.

**Training Details.** All training was conducted on 8 Nvidia H100 GPUs. For specific training hyperparameters of the baseline experiments, please see Appendix 5.1. We utilized the foundational settings consistent with those for DPO (Rafailov et al., 2023) and ORPO (Hong et al., 2024). For our SAPO method, the maximum length for prompts was set at 1792, with the total maximum length for prompts and responses capped at 2048. Training was carried out over four epochs. The segment length for teacher-forcing supervision was 256, the replay buffer was sized at 2000. For each combination of prompt and chosen response, we sampled a single corresponding rejected response. The update coefficient $\alpha$ for the EMA model was set to 0.5. The EMA model was updated every two steps during our training process.

## 5.2 BENCHMARK PERFORMANCE

In our comprehensive analysis of the SAPO framework, detailed in Tables 1 and 2, we systematically evaluate and compare the performance of various models, focusing on multiple dimensions crucial for the performance of LLMs. Table 1 presents a detailed comparative performance analysis on the Open LLM Leaderboard benchmark across various tasks tailored to assess reasoning, language understanding, and problem-solving capabilities. Notably, under DPO and ORPO-based algorithms, SAPO implemented with LLaMA and Mistral architectures, demonstrate superior performance across most datasets, achieving higher average scores. The improvement is particularly notable in the LLaMA models, with the ORPO-enhanced LLaMA reaching an average score of 67.36.

Table 2 evaluates language model alignment performance on benchmarks such as instruction-following (IFEval) and conversational ability (MT-Bench, AlpacaEval 2.0). Here, SAPO achieves high scores in IFEval, indicating its exceptional capability in instruction-following. Additionally, in assessing conversational ability, particularly for the two-turn conversation benchmark MT-Bench, we present scores for each turn and their average score evaluted by GPT-4. SAPO consistently outperforms other models across most settings in MT-Bench, achieving an average score of 7.45 in the ORPO-based LLaMA setting, which highlights its robust multi-turn conversational capabilities. Furthermore, in AlpacaEval 2.0, we present both the length control win rate and the base win rate without length control, alongside the average response length. This data highlights how performance in AlpacaEval is influenced by response length. While SAPO underperforms compared to the LLaMA-based SPIN in single-turn tasks, it shows better performance on the Mistral model. It is important to note that our training was conducted on the multi-turn conversation dataset Distilabel-Capybara, and since AlpacaEval 2.0 primarily consists of single-turn dialogue tasks, we suggest using MT-Bench as a more appropriate metric for evaluating the model's conversational abilities.

## 5.3 ABLATION STUDY

In Table 3, our ablation study of the LLaMA-3-8B model using the ORPO algorithm showed that on-policy sampling led to notable declines in performance on benchmarks like IFEval and the Open

Table 1: Open LLM Leaderboard Evaluation.

| Cases | Method | Arc Challenge | TruthfulQA | Winogrande | GSM8k | Hellaswag | MMLU | Average |
|---|---|---|---|---|---|---|---|---|
| **ORPO-Based** | meta-llama/Meta-Llama-3-8B | 58.02 | 43.92 | 77.43 | 51.48 | 82.10 | 65.13 | 63.01 |
| | ORPO-Llama-3-8B | 60.41 | 57.69 | 77.9 | 55.88 | 82.62 | **64.93** | 66.57 |
| | SPIN-ORPO-Llama-3-8B-Iter3 | 61.09 | 56.87 | 75.22 | 50.80 | **84.31** | 63.12 | 65.24 |
| | SAPO-ORPO-Llama-3-8B | **61.95** | **59.00** | **79.08** | **56.33** | 83.48 | 64.31 | **67.36** |
| | mistralai/Mistral-7B-v0.1 | 61.52 | 42.58 | 77.58 | 37.53 | 83.44 | **62.36** | 60.84 |
| | ORPO-Mistral-7B | 62.80 | 54.41 | 77.90 | 45.26 | 84.16 | 60.82 | 64.23 |
| | SPIN-ORPO-Mistral-7B-Iter3 | 56.48 | 52.91 | 70.80 | 39.88 | 77.65 | 59.35 | 59.51 |
| | SAPO-ORPO-Mistral-7B | **63.14** | **55.00** | **79.16** | **46.70** | **85.02** | 61.42 | **65.07** |
| **DPO-Based** | Meta-Llama-3-8B-SFT | 54.86 | 51.73 | 76.72 | 44.81 | 81.01 | 63.57 | 63.95 |
| | DPO-Llama-3-8B | 57.00 | 53.58 | 77.11 | 46.25 | 81.81 | **63.82** | 63.26 |
| | SPIN-DPO-Llama-3-8B-Iter3 | 56.40 | **55.80** | 77.98 | 50.34 | 82.06 | 63.73 | 64.38 |
| | SAPO-DPO-Llama-3-8B | **57.76** | 55.65 | **78.85** | **52.39** | **82.83** | 63.75 | **65.21** |
| | wandb/mistral-7b-zephyr-sft | 62.63 | 54.00 | 76.32 | 44.88 | 84.77 | 60.93 | 63.92 |
| | DPO-Mistral-7B | 63.14 | 55.81 | 75.69 | 41.02 | 85.16 | **60.97** | 63.63 |
| | SPIN-DPO-Mistral-7B-Iter3 | **64.42** | 55.46 | **76.95** | 44.66 | 85.00 | 60.93 | 64.57 |
| | SAPO-DPO-Mistral-7B | 63.99 | **57.47** | 76.32 | **45.11** | **85.42** | 59.79 | **64.68** |

Table 2: Evaluation on IFEval, MT-Bench, AlpacaEval 2.0.

| Cases | Method | IFEval | MT Bench | | | AlpacaEval 2.0 | | |
|---|---|---|---|---|---|---|---|---|
| | | | First Turn | Second Turn | Average | LC Win-Rate | Win-Rate | Length |
| **ORPO-Based** | ORPO-Llama-3-8B | 49.69 | 7.58 | **7.16** | 7.37 | 9.65 | 8.11 | 1599 |
| | SPIN-ORPO-Llama-3-8B-Iter3 | 48.34 | 7.38 | 6.54 | 6.96 | **14.85** | **13.58** | 1725 |
| | SAPO-ORPO-Llama-3-8B | **50.39** | **7.76** | 7.14 | **7.45** | 9.72 | 8.37 | 1507 |
| | ORPO-Mistral-7B | **57.78** | **7.52** | 6.81 | **7.17** | 13.63 | 10.44 | 1358 |
| | SPIN-ORPO-Mistral-7B-Iter3 | 44.68 | 7.17 | 6.51 | 6.84 | 12.55 | 11.11 | 1610 |
| | SAPO-ORPO-Mistral-7B | 57.60 | 7.43 | **6.86** | 7.15 | **15.56** | **11.41** | 1333 |
| **DPO-Based** | Meta-Llama-3-8B-SFT | 41.43 | 7.12 | 6.48 | 6.80 | 7.22 | 5.27 | 1200 |
| | DPO-Llama-3-8B | 45.50 | 7.40 | 6.68 | 7.04 | 8.94 | 6.67 | 1246 |
| | SPIN-DPO-Llama-3-8B-Iter3 | 45.90 | **7.66** | 6.99 | 7.32 | **10.35** | **13.19** | 2289 |
| | SAPO-DPO-Llama-3-8B | **48.28** | 7.61 | **7.16** | **7.38** | 9.73 | 9.66 | 1833 |
| | wandb/mistral-7b-zephyr-sft | 35.35 | 7.47 | 6.65 | 7.06 | 5.47 | **21.94** | 8193 |
| | DPO-Mistral-7B | 35.33 | 7.57 | 6.78 | 7.18 | 5.41 | 21.17 | 7937 |
| | SPIN-DPO-Mistral-7B-Iter3 | 38.65 | 7.25 | 6.75 | 7.00 | 4.15 | 8.58 | 6683 |
| | SAPO-DPO-Mistral-7B | **44.60** | **7.72** | **7.04** | **7.38** | **11.20** | 15.08 | 2789 |

LLM Leaderboard, which test instruction-following and question-answering capabilities, respectively. This underperformance is likely due to the inherent volatility of on-policy sampling, where rapid shifts in model parameters and fluctuations in training data contribute to inconsistent training outcomes. Conversely, our off-policy strategy using an EMA model with a replay buffer produces more stable and representative data, especially useful when the policy model frequently updates. This approach prevents deviations in behavior that could arise from sampling with an unstable policy model, enhancing training consistency. Meanwhile, generating rejected responses from scratch demonstrated improved performance on IFEval, as it was not constrained by previously chosen responses and could more freely align with the prompt's instructions. However, this approach underperformed on other benchmarks, indicating that completely unrestricted generation may yield lower-quality responses that negatively impact the model's abilities in question-answering and dialogue tasks. Overall, our approach achieved better results across a range of metrics, validating the effectiveness of our training paradigm in promoting stable and consistent responses.

Table 4 presents an ablation on reference model updating strategies based on DPO for LLaMA and Mistral models. It evaluates three approaches: fixed reference (fix-ref), where the reference model remains static; policy reference (policy-ref), updated at intervals with the current policy's weights; and EMA reference (ema-ref), updated periodically with weights from an EMA model. The results highlight the importance of regularly updating the reference model; if the reference model remains unchanged, the learned model might be overly regularized towards the initial SFT model, potentially degrading performance on more complex tasks. The ema-ref update strategy shows the best performance, indicating that smoother updates significantly enhance model stability during training. For our DPO-based experiments, we implemented the ema-ref update strategy.

The ablation study shown in Figure 2 demonstrates how varying training epochs impact the LLaMA-3-8B model's performance under the ORPO algorithm across different benchmarks. As epochs increases, Open LLM Leaderboard scores decrease, signaling potential overfitting and aligning with the "alignment tax" phenomenon (Askell et al., 2021) where excessive alignment with human

Table 3: Ablation of training paradigm.

|  | Open LLM LeaderBoard | IFEval | MT-Bench |
|---|---|---|---|
| on-policy | 65.97 | 36.73 | **7.49** |
| no segment | 67.18 | **52.00** | 7.32 |
| Ours | **67.36** | 50.39 | 7.45 |

Table 4: Ablation of Reference Model Update Strategies.

| Model | Variant | Open LLM LeaderBoard | IFEval |
|---|---|---|---|
| LLaMA | fix-ref | 64.45 | 44.34 |
|  | policy-ref | 64.81 | 47.05 |
|  | ema-ref | **65.21** | **48.28** |
| Mistral | fix-ref | 64.50 | 41.14 |
|  | policy-ref | **64.77** | 41.47 |
|  | ema-ref | 64.68 | **44.60** |

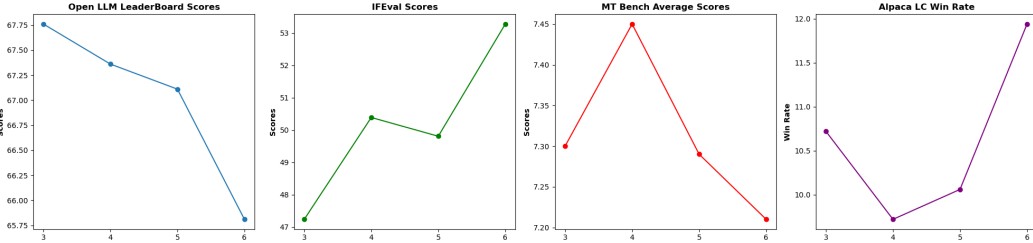

Figure 2: Ablation of training epochs on LLaMA-3-8B using ORPO across multiple benchmarks.

preferences can harm general question-answering abilities. Conversely, IFEval scores improve, indicating enhanced instruction-following skills. MT-Bench scores peak at four epochs, suggesting this as the optimal training length to prevent overfitting. Notably, the Alpaca LC Win Rate also climbs, particularly after the fifth epoch. Given that we are training on the Distilabel-Capybara multi-turn dialogue dataset, our primary focus is on the MT-Bench metrics. After considering the overall performance across various benchmarks, we set the training epoch to 4.

Table 5 showcases an ablation study examining how varying the number of rejected responses per prompt and the segment token length affects model performance, specifically focusing on the LLaMA model using the Odds Ratio Preference Optimization (ORPO) algorithm. Increasing the number of rejected responses from 1 to 4 and then to 8 demonstrates a general trend of reduced performance across most benchmarks like the Open LLM Leaderboard and IFEval, suggesting that a higher number of negative samples might lead to potential overfitting. On the other hand, increasing the segment token length from 64 to 256 enhances model performance, indicating that 256 tokens provide the optimal contextual information for our model.

Table 5: Ablation of the number of rejected responses per prompt and segment token length.

| ORPO Ablation | Value | Open LLM Leaderboard | IFEval | MT Bench | | | AlpacaEval 2.0 | | |
|---|---|---|---|---|---|---|---|---|---|
|  |  |  |  | First Turn | Second Turn | Avg | LC WR | WR | Avg Length |
| Response num per prompt | 1 | 67.36 | 50.39 | 7.76 | 7.14 | 7.45 | 9.72 | 8.37 | 1507 |
|  | 4 | 67.20 | 47.50 | 7.48 | 7.0 | 7.24 | 10.93 | 9.57 | 1595 |
|  | 8 | 66.95 | 47.97 | 7.56 | 7.09 | 7.33 | 10.00 | 8.92 | 1571 |
| Segment token length | 64 | 66.70 | 51.67 | 7.58 | 6.70 | 7.14 | 10.81 | 8.90 | 1521 |
|  | 128 | 67.05 | 50.93 | 7.48 | 6.81 | 7.15 | 9.13 | 8.26 | 1557 |
|  | 256 | 67.36 | 50.39 | 7.76 | 7.14 | 7.45 | 9.72 | 8.37 | 1507 |

## 6 CONLUSION

In this paper, we introduce the Self-Augmented Preference Optimization (SAPO) framework, a dynamic off-policy learning paradigm that updates training data in real-time. Leveraging an Exponential Moving Average (EMA) model and a replay buffer, SAPO ensures stable and consistent performance, drastically reducing dependence on large pre-collected datasets. Through extensive evaluations across diverse Large Language Model (LLM) architectures such as LLaMA-3-8B and Mistral-7B, and using contrastive preference learning algorithms like DPO and ORPO, our method demonstrates superior performance on benchmarks including the Open LLM Leaderboard, IFEval, MT-Bench, and AlpacaEval 2.0. Furthermore, our method's independence from annotated paired data and freedom from iterative training as seen in SPIN, positions it for broader applicability in diverse large-scale post-training tasks, pointing towards promising future directions.

ETHICS STATEMENT

The Self-Augmented Preference Optimization (SAPO) framework improves the alignment of Large Language Models (LLMs) with human values by dynamically updating training data, which reduces dependence on large datasets and helps mitigate potential biases. The use of public datasets enhances transparency and compliance with ethical standards, fostering responsible AI practices. However, it is crucial to recognize that there is still a possibility of generating sensitive or harmful content. Therefore, continuous evaluation and monitoring are imperative to ensure that the deployment of SAPO adheres to ethical AI usage principles.

REPRODUCIBILITY STATEMENT

To facilitate reproducibility, we elaborate our method in Section 3 and provide a comprehensive algorithm box in Algorithm 1. We provide details in method implementation and experimental setups in Section 5.1. Our code can be viewed at https://anonymous.4open.science/r/sapo_review-FDBE.

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

# Appendix

## CONTENTS

## A  LIMITATIONS AND FUTURE WORK

While the Self-Augmented Preference Optimization (SAPO) framework improves LLM alignment with human preferences via dynamic data updates, its reliance on a static, human-generated data distribution limits performance to human levels. For future work, since SAPO does not require paired data, applying the algorithm to large-scale, high-quality SFT datasets for fine-tuning presents a promising direction. Additionally, while current efforts primarily focus on general instruction-following tasks, exploring the application of SAPO to specific domains such as mathematics, reasoning, and code generation offers potential for further advancements.

## B CHAT TEMPLATE

For DPO Rafailov et al. (2023)-based algorithms and LLaMA models, we consistently utilize the ChatML chat template. To align with the default settings of ORPO Hong et al. (2024), we employ the Vicuna chat template Zheng et al. (2024) for ORPO-based Mistral models, and we have omitted system messages.

**ChatML Chat Template Example**

---

<|im_start|>user

How are you today?<|im_end|>

<|im_start|>assistant

I'm doing great, thanks for asking! How can I assist you today?<|im_end|>

<|im_start|>user

Can you help me find a recipe for lasagna?<|im_end|>

<|im_start|>assistant

Absolutely! I can suggest a traditional recipe or a vegetarian option. Which one would you prefer?<|im_end|>

<|im_start|>user

Let's go with the traditional recipe.<|im_end|>

<|im_start|>assistant

Great choice! Here's a simple and delicious traditional lasagna recipe you can try. Would you like the recipe for a specific number of servings?<|im_end|>

---

**Vicuna Chat Template Example**

---

<|user|>

How are you today?

<|assistant|>

I'm doing great, thanks for asking! How can I assist you today?

<|user|>

Can you help me find a recipe for lasagna?

<|assistant|>

Absolutely! I can suggest a traditional recipe or a vegetarian option. Which one would you prefer?

<|user|>

Let's go with the traditional 

<|assistant|>

Great choice! Here's a simple and delicious traditional lasagna recipe you can try. Would you like the recipe for a specific number of servings?

---

