# OpenReview forum: "Self-Augmented Preference Optimization: Off-Policy Paradigms for Language Model Alignment"
_ICLR.cc/2025/Conference — Submitted to ICLR 2025_

### Official Review · Reviewer_QCNp · 2024-10-19

**Soundness:** 2
**Presentation:** 3
**Contribution:** 2
**Rating:** 5
**Confidence:** 4

**Summary:**

This paper proposed a new method, SAPO, for aligning language models with human preferences. Previous methods either requires labeled pairwise data or an external reward signal provider. SAPO overcome this by constructing negative responses from SFT dataset and then adapt DPO / ORPO to train the model. Experiments are conducted to empirically verify the performance of SAPO

**Strengths:**

The strengths of this paper are listed as follows

1. The paper proposes a very novel and interesting way to construct negative responses from high-quality dataset

2. The experiments are conducted over a wide range of models and evaluation benchmarks, which makes it solid and comprehensive

3. The idea of leveraging of EMA and replay buffer is interesting.

**Weaknesses:**

My first concern pertains to the high-level intuition behind the methodology. SAPO requires to generate negative samples to pair up those targets in SFT dataset. However, the motivation behind this is unclear. Specifically, SAPO generates negative responses by replacing segments of the SFT response with the model's own generation, which produces off-policy negative responses. The the DPO / ORPO loss will push the model to further penalize these responses. My concern is that, given that these responses is already unlikely generated by the model, what is the necessity of penalizing them further? What specific benefit does penalizing such off-policy data offer, especially when their likelihood of occurrence is already minimal?

Additionally, while SAPO introduces several improvements over SPIN—including a new method for negative sample generation, as well as the use of EMA and a replay buffer—these enhancements can be integrated into SPIN. However, whether SPIN can benefits from these techniques and how SPIN with these compares to SAPO, both at a high level and empirically, is not discussed sufficiently.

Finally, since SAPO's objective is ultimately to fit the SFT dataset responses, I believe that SFT itself should also be included as a baseline for comparison.

**Questions:**

Did the author conduct any experiments to show that the generated $y^{-}$ is indeed worse than $y^{+}$? For example, you might use some off-the-shelf reward model to compute the average rewards of $y^{+}$ and $y^{-}$.

---

### Official Review · Reviewer_K6Gi · 2024-10-27

**Soundness:** 3
**Presentation:** 3
**Contribution:** 2
**Rating:** 5
**Confidence:** 4

**Summary:**

The authors introduce off-policy reinforcement learning (RL) techniques, including Exponential Moving Average (EMA) and replay buffer, into the self-augmented preference optimization, where only a dataset of desired responses is needed, while negative samples are generated by the model itself during optimization. The proposed approach aims to reuse timely feedback to train the model, thereby mitigating the delays often encountered in the traditional self-play training paradigm, which has separate sampling and training phases. Additionally, the authors propose a segment-level data augmentation strategy, where a segment of the full response is regenerated by the EMA model.

**Strengths:**

The motivation to integrate EMA and replay buffer into the self-play pipeline is well-grounded. Experimental results across various benchmarks demonstrate the proposed method's effectiveness over contrastive and self-play baselines.

**Weaknesses:**

1. One crucial aspect of the self-play policy optimization paradigm is to construct negative samples that are generalizable. However, the segment-level supervision employed by the authors could lead to noticeable discontinuities due to the concatenation of the regenerated segment B' with the original segment C. This may introduce unintended bias in the preference learning stage, which could affect the model's ability to generalize.

2. The paper lacks a direct comparison between the effect of curriculum learning and the approach of SPIN that alternates sampling and training phases. For instance, including a comparison with the SPIN method in Figure 2 could provide clearer insights into the effectiveness of this approach.

3. The ablation study in Table 3 presents inconsistent results across different benchmarks. The proposed segment-level augmentation does not consistently outperform direct sampling, at least based on the current experimental results. Additional studies or further refinement of the augmentation technique could help clarify its benefits, e.g., investigating how varying the segment length affects performance across different benchmarks, or exploring alternative segmentation strategies that might lead to more consistent improvements.

**Questions:**

1. How do the authors address the potential discontinuities caused by segment-level supervision, and what impact do they anticipate these discontinuities might have on the overall model performance and generalizability?

2. In the ablation study (Table 3), the segment-level augmentation shows varying effects on different benchmarks. Have the authors explored any adjustments to this augmentation strategy to mitigate these inconsistencies, or are there alternative augmentation methods they would consider?

3. What is the rationale behind starting with epoch 3 rather than epoch 1 in Figure 2?

4. Can the authors include results for SPIN in Figure 2 to strengthen the comparative analysis and provide more context regarding the effectiveness of their method?

---

### Official Review · Reviewer_HYNW · 2024-11-03

**Soundness:** 2
**Presentation:** 2
**Contribution:** 2
**Rating:** 3
**Confidence:** 4

**Summary:**

The paper presents SAPO, an LLM alignment framework applicable to pairwise preference-based methods like DPO and ORPO with a gradually updated reference model and self-augmented preference pairs. Mistral and Llama models trained with the SAPO framework outperformed conventional offline alignment schemes on both instruction-following benchmarks and leaderboard benchmarks.

**Strengths:**

1. SAPO introduces a versatile LLM alignment framework that can be used in a single trajectory setting, widening the applicability of alignment methods to diverse tasks.
2. SAPO demonstrates strong performance in general compared to the default settings of DPO and ORPO.
3. The paper presents ablations over various experimental axes, providing a better understanding of SAPO's mechanism.

**Weaknesses:**

**1. Impact of EMA and segment-level augmentation in SAPO**

SAPO comprises two main techniques that differ from the conventional offline methods (DPO, ORPO): EMA and segment-level response augmentation. While the method is widely tested over different models and methods, the impact of EMA strategies and hyperparameter choices has not been sufficiently studied. For example, the update coefficient $\alpha$ and EMA update frequency were fixed to 0.5 and 2 throughout the experiments. Also, the impact of segment-level augmentation is partially studied, and only the high-level interpretations were presented in Section 5, lacking an in-depth understanding of how and why it makes SAPO a strong method (which is connected to the questions below). Thus, the necessity of EMA and segment-level augmentation in SAPO is left unclear.

&nbsp;

**2. Clarity in experiments and ablations**

Some explanations are not clear enough to follow. For instance, in the first paragraph of Section 5.3, the experiments on "on-policy sampling" are not clear if means: (1) sampling from the trained policy in the middle of training [1] is assumed as rejected samples, (2) sampling multiple responses from the policy that the trained policy is initialized from and somehow labeled as chosen/rejected [2], or else. Also, it is unclear if "epoch" in the training details of SAPO and "iteration" in Algorithm 1 are equivalent or distinct values. Including these two examples, overall, the paper does not precisely state some terminologies.

&nbsp;
&nbsp;


**References**

[1] Direct Language Model Alignment from Online AI Feedback (Guo et al., 2024)

[2] SimPO: Simple Preference Optimization with a Reference-Free Reward (Meng et al., 2024)

**Questions:**

Along with some points above, some additional questions are:

**1. Clarification on segment-level augmentation?**

As the authors stated in Equations (4) and (5), segment B is selected as a target to refine/augment as a rejected response. Then, the segment C is appended to the refined B' for the next iteration. While refining B and C simultaneously as a continuation of A sounds intuitive, the rationale behind appending C to B' is quite unclear as some contextual mismatch could exist between B' and C. What are the intuitions behind the choice of this specific segmentation scheme?


&nbsp;


**2. Segment token length and general performance?**

The last paragraph of Section 5.3 ablates different segment token lengths and concludes that 256 was ideal. However, the impact of 256 tokens differs by the expected response length of the dataset, especially in the general chat dataset like Capybara. Interpreting the effect of segment token length by its ratio over the expected response length would more clearly demonstrate the effectiveness of SAPO.


&nbsp;


**3. Abnormal AlpacaEval generation length for Mistral-SFT in Table 2?**

I noticed that the AlpacaEval 2.0 generation length for wandb/mistral-7b-zephyr-sft is excessively long, as are the DPO and SPIN-DPO trained versions. However, SAPO-DPO-Mistral-7B suddenly recovers to the normal range (referred to the overall generation lengths of the models registered in the official leaderboard). Some clarifications on the excessively long generation length of wandb/mistral-7b-zephyr-sft and some insights on how SAPO-DPO is the only method resolving such issue in the Table would be helpful.

---

### Official Review · Reviewer_kaVT · 2024-11-11

**Soundness:** 2
**Presentation:** 3
**Contribution:** 2
**Rating:** 3
**Confidence:** 4

**Summary:**

The paper introduces an improved method for LLM alignment called SAPO. The method targets on mitigating the need for paired preference data in the alignment stage, by introducing EMA model and data augmentation methods for creating dispreference data. Results show that SAPO achieves some improvements compared to selected baselines.

**Strengths:**

1. Abundant Experiments: The paper conducts many experiments to show the performances of the proposed and compared methods
2. Clear Presentation: The paper's presentation is clear and easy to follow

**Weaknesses:**

1. Marginal and Inconsistent Improvement: Experiments show that the improvements from SAPO in many cases are marginal (e.g., DPO-based Llama-3-8B improves 0.8 and Mistral-7B improves 0.1 from the SPIN baseline). Sometimes, it is even worse than baseline methods (e.g., AlpacaEval 2.0).
2. Potential Low Training Efficiency: Since the method involves sampling and building new dispreferred responses at each iteration, its training efficiency could be problematic compared to typical self-play method like SPIN which samples at the end of each phase. Can you provide detailed profiling of the time consumed for each stage in the iteration?
3. Weak Baselines: I notice authors to use meta-llama/Meta-Llama-3-8B and mistralai/Mistral-7B-v0.1 (which are the base models, rather than instruct ones) as the baselines. However, these base models have not been aligned using SFT and RLHF to serve as proper baselines against those alignment methods.
4. Intuition of generating B': I am concerned with the continuity of regenerating middle segment B' from B. How do you select the segment,  by mere 256 tokens? In that case, how to ensure that the new B' is semantically continuous to the original C? Besides, it is doubtful that the regenerated B' is guaranteed to be worse than the original one. These questions make me unconfident of the reasonability of the method.

**Questions:**

See above questions.

---

### Meta-Review · Area_Chair_SdU8 · 2024-12-19

**Metareview:**

Summary: This paper pointed out that LLM alignment methods like DPO are limited to static preference data. This paper proposed Self-Augmented Preference Optimization (SAPO) to autonomously generate negative responses without prepared paired preference data. Experiments conducted on LLaMA3-8B and Mistral-7B models across several benchmarks demonstrate the effectiveness compared to offline baselines and self-play methods.

Strengths: (1) The model is tested across several benchmarks and base models. (2) The idea of constructing negative responses from high-quality dataset is interesting.

Weaknesses: (1) The improvements of SAPO compared to baselines are marginal, and some baselines are weak as they are not alignment methods. (2) The impact of the segment-level supervision is not clear. (3) The motivation of SAPO is not clear enough.

This paper received four negative ratings as final recommendations, i.e., 5, 5, 3, 3. The authors did not respond to reviewers' comment. The AC agree with reviewers on the reject decision.

**Additional Comments On Reviewer Discussion:**

The authors did not respond to the reviewers' comments.

---

### Decision · Program_Chairs · 2025-01-22

Reject